# Investigations on the Band-Gap Characteristics of Variable Cross-Section Periodic Structure Support Made of Acrylonitrile-Butadiene-Styrene

**DOI:** 10.3390/ma15124308

**Published:** 2022-06-17

**Authors:** Jinguang Zhang, Xu Xia, Xianglong Wen, Meng Zang, Yukuan Dou

**Affiliations:** 1School of Mechanical and Electronic Engineering, Wuhan University of Technology, Wuhan 430070, China; jgzhang@whut.edu.cn (J.Z.); xiaxu@whut.edu.cn (X.X.); hj2528g@163.com (M.Z.); xx76jj@163.com (Y.D.); 2Institute of Advanced Material and Manufacturing Technology, Wuhan University of Technology, Wuhan 430070, China; 3Hubei Provincial Engineering Technology Research Center for Magnetic Suspension, Wuhan 430070, China; 4Hainan Special PhD Scientific Research Foundation of Sanya Yazhou Bay Science and Technology City, Sanya 572024, China

**Keywords:** periodic structure support, variable cross-section, band gap characteristics, vibration

## Abstract

Based on the band gap theory of periodic structure, this article proposes a new variable cross-section periodic structure support made of acrylonitrile-butadiene-styrene. The band gap characteristics of the periodic structure support were studied experimentally. According to the basic theory of band gap calculation, two kinds of supports with the same installation size were designed, and they were manufactured by 3D printer. Then, the displacement–load curve and the vibration characteristic curves of the periodic structure support were obtained through simulation analysis. The band gap range of the two supports was measured by hammer excitation, and the accuracy of the finite element model was verified by comparison with the experiment results. Finally, the response curve of the periodic structure support with variable cross-section every 100 Hz was obtained by excitation of the shaker, which verified the vibration isolation effect of the measured band gap. The results show a band gap in the support of the variable cross-section periodic structure, compared with the support of the non-periodic structure. If the vibration frequency is within the band gap frequency, the vibration will be significantly attenuated.

## 1. Introduction

When helicopters, ships and other equipment are running, their motors produce a heavy vibration, and the vibration force is transferred to the fuselage through the support, creating a very negative dynamic environment for the whole equipment [1]. Heavy vibrations not only potentially harm operators, but also cause damage to airborne equipment, including structural fatigue damage. Therefore, it is necessary to add vibration isolation equipment between the motor and the fuselage. Vibration isolation design is usually adopted for the transmission path of the motor support structure [2]. In traditional vibration isolation design, in order to obtain a better vibration isolation effect, the vibration isolator usually requires low stiffness, which limits static deformation. Large static deformation stiffness is detrimental to effective work, and should be avoided. In order to solve this problem, active control methods have been adopted. Hoffmann et al. [3] applied control of the narrowband multi-channel filtered-x version of the adaptive LMS algorithm to reduce the noise inside a helicopter. Simon et al. [4] devised an active (or semi-active) control program and applied it to the vibration-acoustic transmission of the isolation panels, then to the anti-torque plates in the helicopter model, and finally in flight to reduce noise from gearbox vibrations. Ma et al. [5] proposed a new discrete predictive sliding mode control scheme based on a controlled autoregressive moving average model. The real-time active control experiments were carried out on the newly developed helicopter model system and verified the effect of the algorithm. Wang et al. [6] proposed a novel method of structure-borne sound analysis and active force control, combining interval mathematics and robust optimization theorems, to achieve vibration damping and noise reduction for enclosed cavity systems with bounded uncertainty. Lang et al. [7] installed a piezoelectric stack actuator on the gearbox support struts and proposed an adaptive harmonic feedforward/sliding mode feedback hybrid active control method for helicopter fuselage vibration, which can effectively control the vibration of the helicopter fuselage. Kim et al. [8] employed algorithms via two nonlinear methods and a model-based approach, and validated the feasibility of these algorithms for narrowband or wideband control through experiments on a single smart pillar. Active control has better vibration isolation effect, but its use is limited by reliability and system robustness. Therefore, the periodic structure was introduced into the passive control vibration isolation structure to replace the solid line with vibration isolation [6].

With the development of science and technology, the vibration isolation characteristics of periodic structures have been studied in depth. Rayleigh studied the propagation of waves in periodic media with the one-dimensional Mathieu equation. Bloch generalized Floquet’s conclusion to two dimensions and obtained the famous Bloch theorem [9,10]. Refs. [11,12,13] studied the vibration isolation performance of beam-like structures of periodic structures. Refs. [14,15] studied the effect of damping on the band gap of periodic structures. Refs. [16,17] found that the band gap can be tuned by adjusting the geometric parameter values. Refs. [18,19] studied the vibration characteristics of beams with additional inertia and variable cross-section beams. Ref. [20] utilized the modal coupling of a lever mass and a conventional local resonant mass connected separately to the base to form a double-attenuation bandgap. Duhamel et al. [21] proposed a finite element program to calculate the dynamic response of infinite periodic structures under local time-dependent excitation.

The above research shows that the band gap characteristics of periodic structures have vibration attenuation characteristics, so the band gap characteristics of periodic structures can be taken into account when designing support structures.

In recent years, vibration and noise suppression of support structures based on periodic design has attracted attention due to its unique advantages. Szefi and Asiri et al. [22,23] embedded the metal/rubber periodic structure into the main vibration isolator, and realized the adjustment of the structural stopband frequency range through parameter optimization, verifying the effectiveness of the method. After considering the safety of rubber materials in engineering applications, Yang et al. designed a rubber/metal series-parallel composite periodic structure support base and verified the vibration isolation effect of the structure through theory and experiments. A metal structure was added to ensure its reliability [24,25]. Wang et al. designed a composite periodic structure support seat, and verified the vibration attenuation effect of the structure in the axial and lateral directions through experiments [26]; using the genetic algorithm to optimize the design of the composite periodic structure, the attenuation structure optimization of 26.22 dB was achieved [27]. They proposed a new geometric discontinuous periodic structure, and experimentally verified the feasibility and effectiveness of its vibration damping effectiveness [2].

Generally, metal supports are only used as connecting structural parts, and their vibration characteristics have not been effectively considered. Considering vibration characteristics, the support is generally made of rubber or metal/rubber material. Due to its low stiffness, rubber material cannot meet the requirements of the mechanism for the stability while providing vibration isolation characteristics. It cannot process the vibration isolation requirements associated with periodic structure materials, and it is difficult to secure to other materials.

This paper focuses on the study of vibration isolation within the support. Based on the band gap theory of periodic structures, a new structure of the support with a variable cross-section periodic structure made of acrylonitrile-butadiene-styrene was designed with the aim of reducing vibration and noise. In this paper, a thin-shell cylindrical variable cross-section periodic structure support is described, and the basic theory of band gap calculation for infinite periodic structures is introduced. Structures were manufactured by 3D printer, and the software ABAQUS was employed to analyze its mechanical properties and band gap characteristics. Then, the vibration characteristics of the two structures were studied by the hammer method, and the band gap range of the periodic structure support was tested. Finally, through excitation using the exciter, the response curve of the periodic structure support at every 100 Hz was obtained, and the vibration characteristics of the periodic structure support were analyzed.

## 2. Theoretical Analysis of Periodic Structure Vibration Band Gap

The periodic structure can be simplified into a common dynamic model; a single periodic structure is considered to be composed of *n* springs, *n* dampers, and *n* masses in series. The stiffness coefficients of the springs are *k*_1_~*k_n_*, the masses are *m*_1_~*m_n_*, and the damping is *c*_1_~*c_n_*, as shown in Figure 1.

Ignoring the influence of damping on the calculation process, assuming that the motion direction is only the direction X, and displacement corresponding to a certain mass element *m_j_* in the period *x_j_*, the single-period structure can be simplified to a series of n springs and n masses. The infinite-period structural dynamics model can be simplified as shown in Figure 2.

The *j*th mass element motion equation can be expressed as:(1)$$m_{j}{\ddot{x}}_{j}=k_{j}\left(x_{j+1}-x_{j}\right)-k_{j-1}\left(x_{j}-x_{j-1}\right)\;j=1,\;2,\;\ldots ,\;n$$

According to the Bloch theorem, it is written as a harmonic motion with amplitude *A_j_* and angular frequency ω [28].
(2)$$x_{j}=A_{j}e^{i(q\sum_{j=1}^{j}d_{j}-\omega t)}$$
where, $q\sum_{j=1}^{j}d_{j}$ represents the phase factor of the *j*th mass element, $d_{j}$ represents the distance between the *j* and *j* + 1 oscillators, and *q* is the wave vector, which is valued at (−π/a, π/a) in the first Brillouin area [29], and $a=q\sum_{j=1}^{n}d_{j}$.

Substituting Equation (2) into Equation (1), after simplification:(3)$$\left(\frac{k_{j}+k_{j-1}}{m_{j}}-\omega^{2}\right)A_{j}=\frac{k_{j}}{m_{j}}e^{iqd}A_{j+1}+\frac{k_{j-1}}{m_{j}}e^{-iqd}A_{j-1}$$

There are boundary conditions within the periodic structure [29]:(4)$$\{\begin{array}{ll} k_{0}=k_{n} & k_{1}=k_{n+1}\\ m_{0}=m_{n} & m_{1}=m_{n+1}\\ A_{0}=A_{n} & A_{1}=A_{n+1} \end{array}$$

Substitute Equation (4) into Equation (3) and express it in matrix form as:(5)$$\left[\begin{array}{cccccc} \frac{k_{1}+k_{n}}{m_{1}} & -\frac{k_{1}}{m_{1}}e^{iqd} & 0 & \cdots & 0 & -\frac{k_{n}}{m_{1}}e^{-iqd}\\ -\frac{k_{1}}{m_{2}}e^{-iqd} & \frac{k_{2}+k_{1}}{m_{2}} & -\frac{k_{2}}{m_{2}}e^{iqd} & 0 & \cdots & 0\\ \vdots & \vdots & \vdots & \vdots & \vdots & \vdots \\ -\frac{k_{n}}{m_{n}}e^{iqd} & 0 & \cdots & 0 & -\frac{k_{n-1}}{m_{n}}e^{-iqd} & \frac{k_{n}+k_{n-1}}{m_{n}} \end{array}\right]\left[\begin{array}{c} A_{1}\\ A_{2}\\ \vdots \\ A_{n} \end{array}\right]=\omega^{2}\left[\begin{array}{c} A_{1}\\ A_{2}\\ \vdots \\ A_{n} \end{array}\right]$$

The eigenvalues of the standard matrix are as follows:(6)$$\left(X(q)-\omega^{2}I\right)A=0$$

In the Equation (6), $A\;=\;{\left[A_{1},A_{2},\cdots ,A_{n}\right]}^{T}$. In Equation (3), if A has a solution different from zero, its coefficient determinant must be equal to zero, and the solution of Equation (6) is transformed into the eigenvalue problem of solving the general matrix *x* of *n* × *n*, from which can be obtained the elastic-band structure of the spring oscillator structure.

## 3. Experimental Program of Band Gap Measurement of Periodic Structure

### 3.1. Design and Manufacture of Specimens

According to Equations (1)–(6), when designing the periodic structure support, it is necessary to consider the number of periods, the unit size in a period and the influence of the structural characteristics in a period on its band gap, so the periodic structure was designed as a two-period variable cross-section support.

Existing supporting structures mainly include metal damping support and rubber damping support. The mass of the metal damping support is large and the support has poor deformation effect, meaning it is difficult to recover after deformation. Rubber support is prone to fatigue and aging so its service life is short, which leads to increased costs and wasted resources.

In this paper, a periodic structure support made of acrylonitrile-butadiene-styrene material was used for the purpose of studying its band gap characteristics. Most acrylonitrile-butadiene-styrene is non-toxic, typically opaque and impermeable, as well as hard and tough, with low water acrylonitrile-butadiene-styrene options. This kind of polymer plastic has the characteristics of oil resistance, corrosion resistance, hard texture, and high rigidity.

The specimens had a total length of 135 mm, including two cells with D1 = 60 mm, D2 = 80 mm, L1 = L2 = 5 mm, H1 = H2 = 135 mm, La = 15 mm, Lb = 5 mm, Lc = 15 mm, and Ld = 5 mm, as shown in Figure 3. Structure A was designed as a periodic structure support with two periods, and structure B as a straight-tube structure support for comparative experiment. The assembly dimensions of the two structures were identical. The specimens were formed by 3D printer, using the printer model UP300D produced by Tiertime, the printing material was acrylonitrile-butadiene-styrene, the material filling rate was 32% when printing, and the performance parameters provided by the company are shown in Table 1. The specimens are shown in Figure 3.

### 3.2. Experimental Program with Hammer Excitation

The acquisition instrument used in the experiment and the block diagram of the testing process are shown in Figure 4. A force hammer 8206-002 was used to generate the excitation signal, and vertical excitation was carried out on the periodic structure support pedestal. Two B & K 4507Bx acceleration sensors obtained the input and output responses of the periodic structure support, and input them into B & K’s data acquisition system. The computer software B & K Connect was applied for the data acquisition and analysis.

The experimental bench set-up is shown in Figure 5. The support was installed on the metal platform with bolts, and the metal platform connected with the ground through the base support. An excitation point was set at the bottom surface of the metal platform, and a force hammer was used to hit the excitation point to produce vibration. Two acceleration sensors were installed on the upper and lower surfaces of the specimens, respectively, to collect the input and output acceleration signals. The signals were transmitted to the data processing system through the data acquisition system. The data processing system was used to read, save and process the data. The specific parameters of the equipment are shown in Table 2. The device picture is shown in Figure 6.

### 3.3. Experimental Program with Exciter Excitation

The experimental program was similar to the previous method, except that the excitation equipment was changed to generate an excitation signal by using Modal Exciter Type 4824, with the exciter driven by a power amplifier to vertically excite the support seat of the periodic structure. The excitation point of the exciter was the bottom surface of the metal platform. The experiment excitation bandwidth was from 100 Hz to 3200 Hz, and it was excited once every 100 Hz. The experiment process block diagram and equipment diagram are shown in Figure 7 and Figure 8.

### 3.4. Stiffness Experiment of Periodic Structure Support

The electronic universal testing machine was used to investigate the stiffness performance of the periodic structure support, and the installation method as shown in Figure 9 was adopted for the experiment. Figure 10 shows the electronic universal testing machine. The experiment type was set as compression experiment on the control computer and the size of the sample was input. The loading direction was downward, the loading rate was 1 mm/min, and the sampling frequency was 100 Hz. When the loading displacement reached 10 mm, the loading stopped. The load-displacement curve of the experiment was exported from the supporting software of the electronic universal testing machine.

The experiment results of the above three experiments were analyzed, and are reported in the results and discussion below. Each sample was tested five times, and the two sets of data with the largest and smallest errors were abandoned. The data shown in the text is the mean of the remaining three sets of data.

## 4. Simulation Analysis of Mechanical Properties and Band Gap Characteristics of Bracing Seat with Periodic Structure

### 4.1. Finite Element Modeling of Periodic Structure Support

Procedural aspects in the present work such as meshing, element type selection, boundary conditions, and loading method were carried out according to [29,30,31,32]. According to the structure shown in Figure 3, a 3D model was created by 3D-modeling software and imported into ABAQUS. The periodic structure mesh type was C3D10, the overall size of the mesh was 3 mm, and it comprised of 187,597 elements [31]. The structural finite element is shown in Figure 11.

### 4.2. Simulation Parameter Settings

The simulation material was the same as the production material, which was acrylonitrile-butadiene-styrene. The material parameters of the simulation calculation are shown in Table 1.

During the mechanical analysis, the general static analysis step (Static, General) was used for calculation. The constraint condition was to fix the bottom end face of the support base, and set a reference point on the top of the support base to couple with the top [32]. A time-varying linear displacement was then applied at the reference point. Finally, the displacement and support reaction force in the numerical direction of the reference point were output, and the displacement load curve was obtained. The specific constraints are shown in Figure 12.

Vibration analysis was performed using modal analysis (Frequency) and random vibration analysis steps (Steady-state dynamics, Direct). The lower surface of the support base was fixed, and random vibration of 1–3200 Hz at the bottom constraint was loaded. The simulation results are discussed in Results and discussion.

## 5. Results and Discussion

### 5.1. Simulation Results of Periodic Structural Support

The simulation results use *FRF* to characterize the vibration isolation effect. The *FRF* of the periodic structure can be obtained by Equation (7) [33].
(7)$$FRF=20\log\frac{a_{2}}{a_{1}}$$
where *a*_1_ and *a*_2_ are the vibration accelerations at the input and output terminals, respectively.

Through the simulation analysis, the load-displacement curve and *FRF* value curve of the periodic structure can be obtained. As shown in Figure 13. From Figure 13a, it can be calculated that the stiffness of the periodic structure support seat is 240.8 N/mm. The following conclusions can be drawn from the *FRF* value of Figure 13b: When the excitation frequency was from 369 Hz to 439 Hz, the vibration attenuation of the periodic structure increased gradually. When the excitation frequency was 439 Hz, the *FRF* value was −15.8 dB. However, when the excitation frequency was 488 Hz to 566 Hz, the vibration attenuation effect was weakened; it was −2.7 dB when the excitation frequency was 566 Hz. When the excitation frequency was greater than 566 Hz, the vibration attenuation began to increase and reached its maximum at 1575 Hz, at which time the *FRF* value was −28.8 dB. At 1034 Hz, the vibration attenuation became smaller, and the *FRF* value was −9.7 dB. It can be observed that the band gap of the periodic structure calculated by simulation was 566 Hz to 1995 Hz.

### 5.2. Hammer Excitation Experimental Results

After completing the experiment with hammer excitation, the time domain signal measured by the acceleration sensor was processed using B & K Connect, the vibration response curves for supports of different structures were acquired. The time domain responses of the input and output signals of the straight-tube structure support are shown in Figure 14.

The time domain responses of the input and output signals of the periodic structure support are shown in Figure 15.

The fast Fourier transform of the collected data was directly performed in the software B & K Connect to obtain the frequency response curves of different structures. The frequency responses of the input and output signals of the straight-tube structure support base are shown in Figure 16, and the frequency responses of the input and output signals of the periodic structure support base are shown in Figure 17.

It can be seen from Figure 16 and Figure 17 that the two structures had different degrees of vibration isolation effect at different excitation frequencies, and the vibration isolation effect of the periodic structure was better and more obvious. When frequency was lower than 700 Hz, both structures showed no vibration isolation effect, and even amplified the vibration.

In order more intuitively to display the vibration isolation effect of the two different structures, the *Z* vibration level transformation may be performed on the obtained results. The calculation formula of the *Z* vibration level *VLz* is [34]:(8)$$VLz=20\log\frac{a}{a_{0}}$$
where, *a* is the measured acceleration value; *a*_0_ is the reference acceleration, according to GB 50894-2013, take $a_{0}=10^{-6}{\;m/s}^{2}$.

The *VL_Z_* values of the input and output responses of the two structures are shown in Figure 18 and Figure 19.

It can be seen from Figure 18 that the straight-tube structure began to show a vibration isolation effect when the frequency reached 700 Hz. When the excitation frequency was 1056–1208 Hz, 1256–1432 Hz, 1728–1848 Hz and 2528–2912 Hz, the vibration attenuation was above 10 dB. Vibration isolation at 1368 Hz was up to 30 dB, and it can be seen from Figure 19 that the periodic structure also had a vibration isolation effect at frequency 700 Hz. When the excitation frequency was 696 Hz to 1752 Hz, the vibration isolation effect was above 10 dB, most of the vibration isolation effects were above 20 dB, and there were three intervals above 30 dB. However, the effect of vibration attenuation was reduced when the frequency was 1008 Hz.

The *FRF* values of the two structures were compared to show the vibration characteristics of the two structures. The curves of the *FRF* value of the straight-tube structure and the periodic structure as a function of frequency are shown in Figure 20. It can be seen that the vibration isolation effects of the two structures were different, and a larger band gap can be observed in the periodic structure. Between 696 Hz and 1752 Hz, the vibration isolation and attenuation effect of the periodic structure support was better than that of the straight cylindrical structure support for most frequencies, and the maximum value of the *FRF* value of the periodic structure was up to 37 dB larger than that of the straight structure. However, at frequency 2600 Hz to 2900 Hz, the vibration isolation performance of the straight-tube structure support was obviously better than that of the periodic structure.

### 5.3. Exciter Excitation Experimental Results

After the vibration excitation experiment using the exciter was completed, the time domain signal measured by the acceleration sensor was read in the software LABSHOP, and the vibration response curve of the support of the periodic structure acquired.

The input signal is a sinusoidal signal. Figure 21 shows the input and output time domain response curves at 1600 Hz. The fast Fourier transform of the acquired data was directly performed in the LABSHOP software, and the frequency responses of the input and output signals of the periodic structure support are given in Figure 22. Calculating the *VL_Z_* value and *FRF* value of the periodic structure support according to the data obtained from the experiment, as shown in Figure 23 and Figure 24, it can be clearly seen from Figure 21 and Figure 22 that when the vibration frequency was 1600 Hz, the vibration underwent a significant attenuation effect. When the vibration exciter was employed to excite the vibration, the excitation frequency was from 800 Hz to 2600 Hz, and the vibration was obviously attenuated, except at 1400 Hz, which proves that the band gap measured above is effective. However, the attenuation from 800 Hz to 2000 Hz was different from that 2000 Hz to 2600 Hz. When the excitation frequency was from 800 Hz to 2000 Hz, the input excitation tended to increase, but the output response decreased, and the maximum attenuation was more than 35 dB. However, when the frequency excitation was from 2000 Hz to 2600 Hz, both the input excitation and the output response showed a narrowing trend, and the maximum attenuation was only 15 dB. Therefore, it can be judged that 800 Hz to 2000 Hz is a band gap of the specimen, and the attenuation from 2000 Hz to 2600 Hz is because of the excitation input attenuation. At frequency 1400 Hz, the vibration isolation effect becomes poor, which may be caused by the resonance of the vibration generator due to the coincidence of the excitation frequency with the natural frequency of the periodic structure support base.

### 5.4. Comparison of Simulation and Experimental Results

The load-displacement curve obtained by simulation was compared with that obtained from the experiment, as shown in Figure 25. It can be seen that the stiffness of the periodic structure support obtained by simulation was 240.8 N/mm. That obtained experimentally was 233.5 N/mm, and the simulation stiffness was linear. The stiffness characteristic curve obtained from the experiment was also approximately linear, and the stiffness error is within 3.1%, which can verify the accuracy of the load–displacement curve obtained from the simulation.

The *FRF* values of the simulation data for the periodic structure support, the data for the hammer excitation experiment, and for the exciter excitation experiment were compared, as shown in Figure 26. The black solid line represents the FRF value of the periodic structure support when excited by the vibration exciter, the red solid line represents the *FRF* value excited by the hammer, and the blue dotted line represents the *FRF* value of the finite element simulation. The simulation results show that the band gap was from 566 Hz to 1995 Hz, the bandwidth was 1429 Hz, the center of the band gap was 1280.5 Hz, and the vibration attenuation became smaller at 1034 Hz. The band gap obtained by hammer excitation was from 696 Hz to 1752 Hz, the bandwidth 1056 Hz, the center of the band gap 1224 Hz, and the position of vibration attenuation 1008 Hz. The bandwidth error of hammer excitation and simulation was 26.1%, and the band gap center error was 17.5%. The vibration attenuation curve for the excitation by the exciter is consistent with that by the hammer, thus the accuracy of the vibration isolation effect in the band gap is verified.

## 6. Conclusions

In this paper, on the basis of the band gap theory of periodic structures, a new variable cross-section periodic structure support made of acrylonitrile-butadiene-styrene was designed. By changing the form of the support from a straight tube structure to a periodic structure, the vibration isolation of the support was realized by utilizing the band gap characteristic of the periodic structure. The main conclusions are as follows:(1)A geometrically discontinuous periodic structure was designed, with a single-material variable section. Through 3D printing technology, this periodic structure support was fabricated using ABS.(2)The finite element model of the periodic support was established, its stiffness and band gap characteristics were simulated; and the displacement load curve and the range of the band gap for the periodic support were obtained.(3)The vibration experiment was carried out on the specimens using hammer excitation, and the band gap range of the periodic structure support was obtained. In the band gap range, the periodic structure support showed more obvious vibration attenuation than the straight-tube structure support, and the maximum attenuation reached more than 35 dB. The bandwidth error between the hammer excitation vibration experiment and simulation was 26.1%, and the bandgap center error was 17.5%. The vibration isolation performance of the obtained periodic structure bandgap was verified by using a vibration exciter in the range of 1 Hz to 3200 Hz.(4)At present, this paper verifies that the periodic structure is applied to the design of the support structure, and that the vibration isolation of the support can be realized by using the band gap characteristic. This conclusion can be extended to the design of various support structures. In future work, the factors affecting the band gap of this structure will be explored, including the size and number of periodic structures, and their material qualities.

## Figures and Tables

**Figure 1 materials-15-04308-f001:**
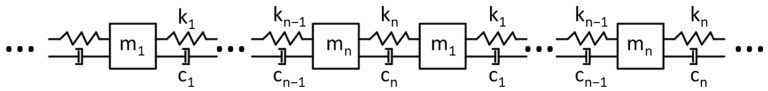
Dynamic Model of Infinite Periodic Structures.

**Figure 2 materials-15-04308-f002:**
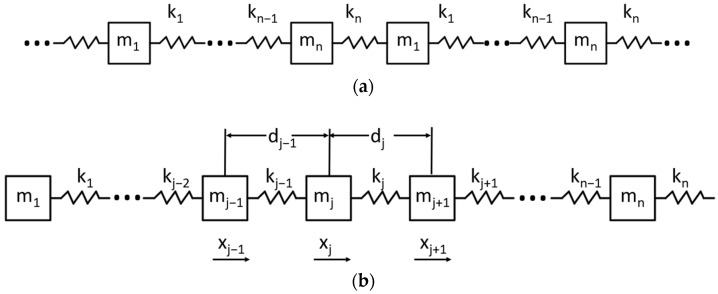
(**a**) Infinite Periodic Elastic Oscillator Structure (**b**) Single Period of Infinite Periodic Spring Oscillator Structure.

**Figure 3 materials-15-04308-f003:**
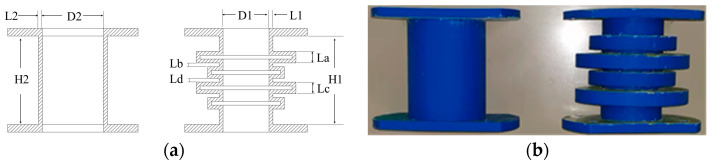
Schematic diagram and reality of specimens. (**a**) Schematic diagram of the structure of the two specimens, (**b**) the two different specimens.

**Figure 4 materials-15-04308-f004:**
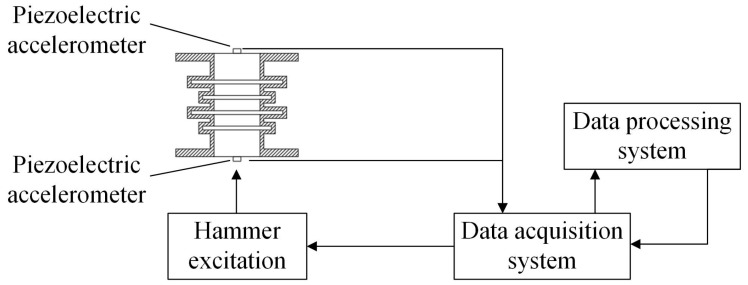
Hammer experiment process block diagram.

**Figure 5 materials-15-04308-f005:**
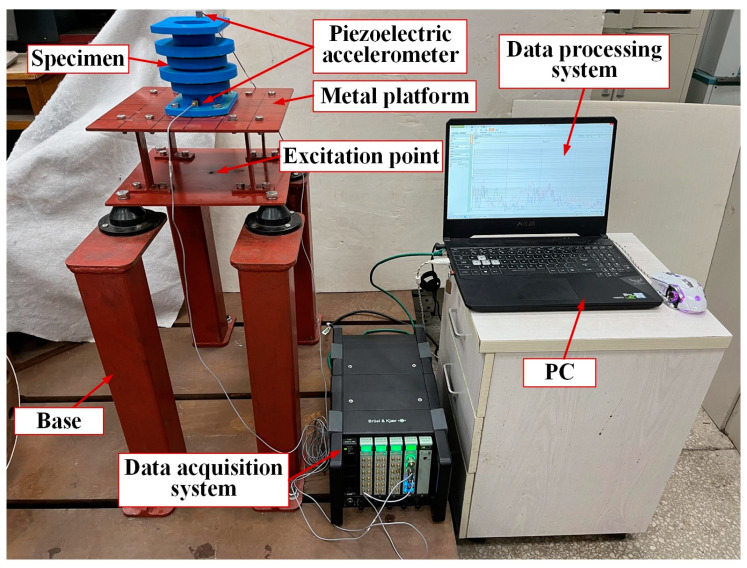
Hammer experiment bench installation diagram.

**Figure 6 materials-15-04308-f006:**
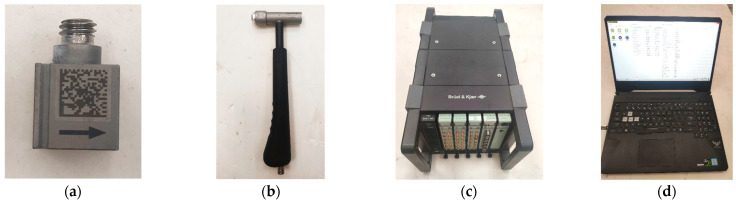
Hammer experiment equipment. (**a**) Piezoelectric accelerometer, (**b**) force hammer, (**c**) data acquisition system, (**d**) data processing system.

**Figure 7 materials-15-04308-f007:**
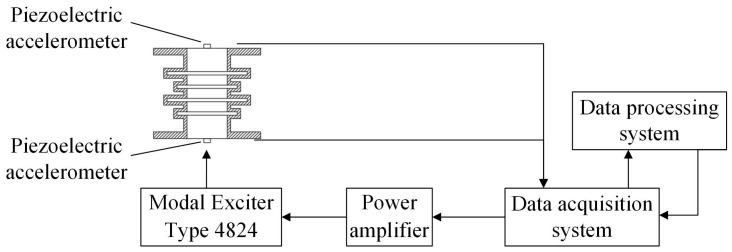
Exciter experiment process block diagram.

**Figure 8 materials-15-04308-f008:**
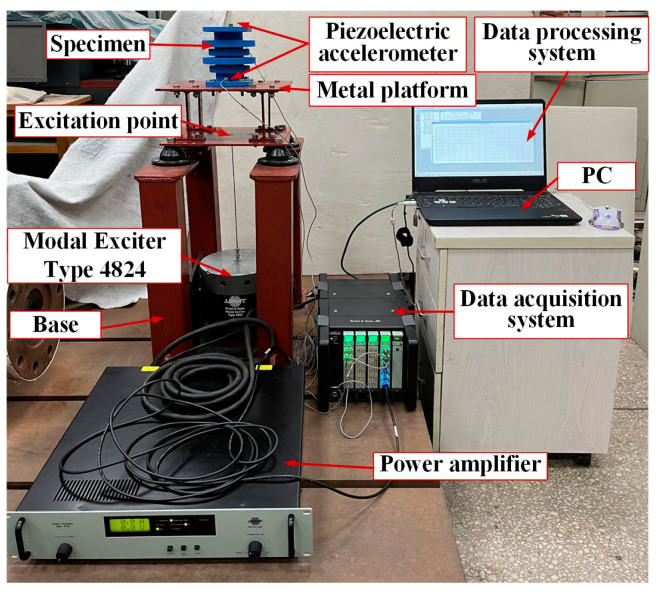
Exciter experiment bench installation diagram.

**Figure 9 materials-15-04308-f009:**
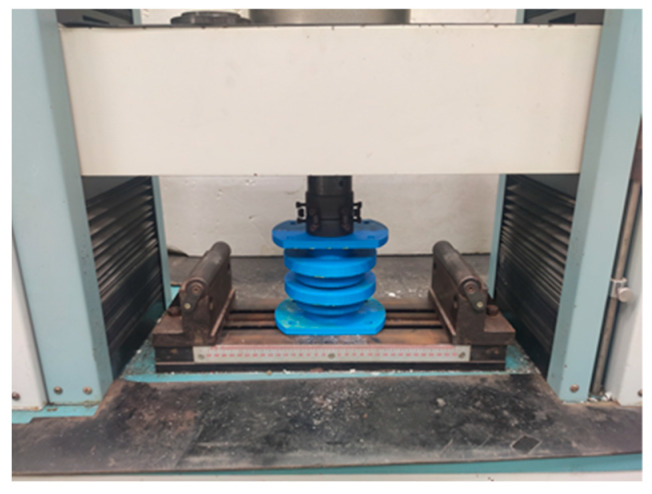
Installation diagram of specimen.

**Figure 10 materials-15-04308-f010:**
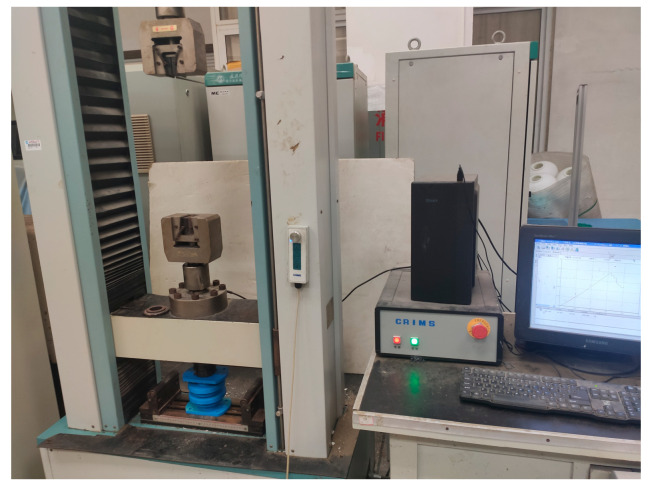
Electronic universal testing machine.

**Figure 11 materials-15-04308-f011:**
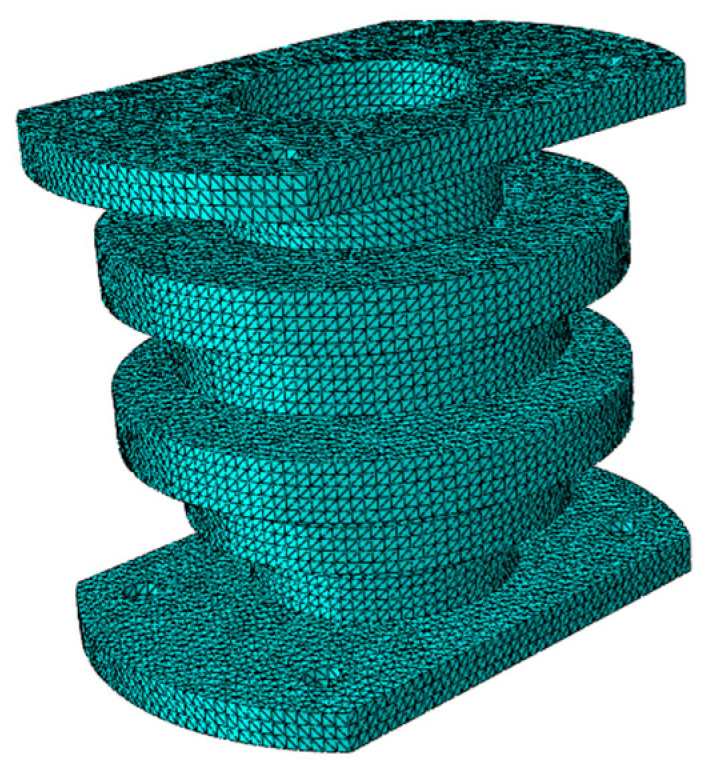
Finite element model of periodic structure support.

**Figure 12 materials-15-04308-f012:**
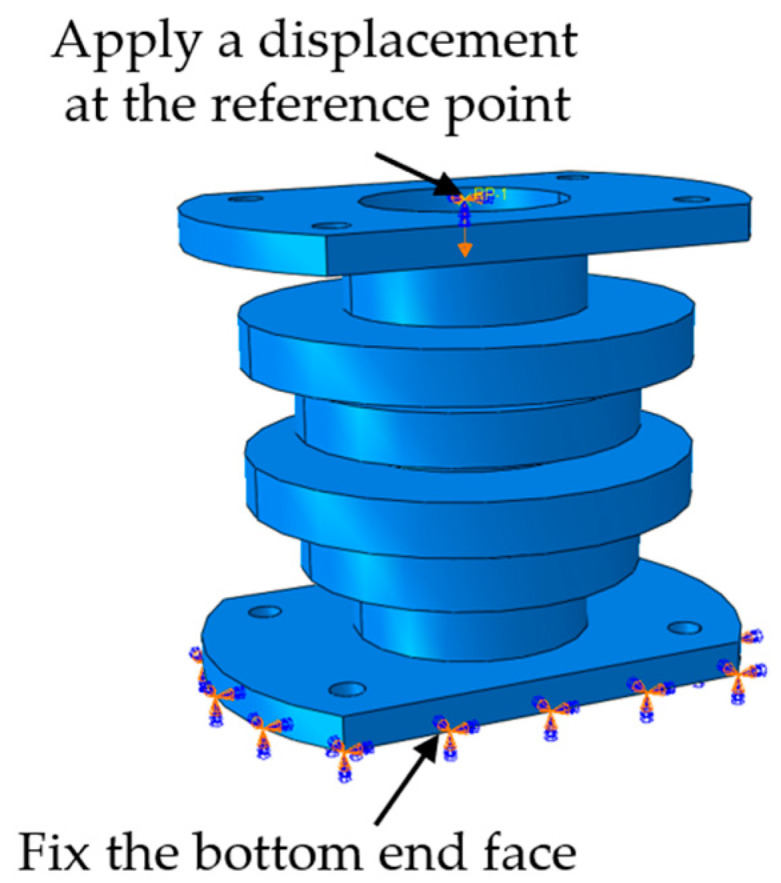
Simulation boundary condition settings.

**Figure 13 materials-15-04308-f013:**
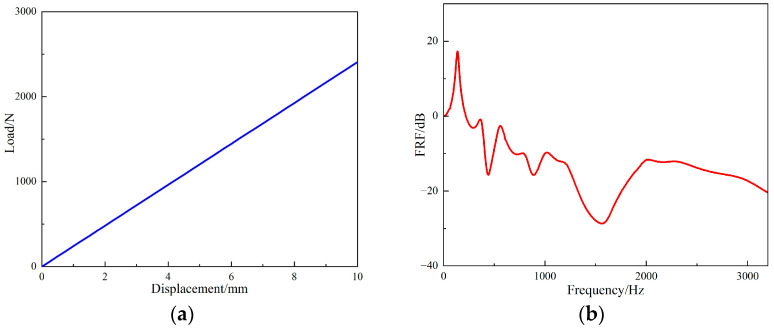
Simulation results of periodic structural support. (**a**) Load-displacement curve of periodic structure support, (**b**) simulation *FRF* value curve of periodic structure support.

**Figure 14 materials-15-04308-f014:**
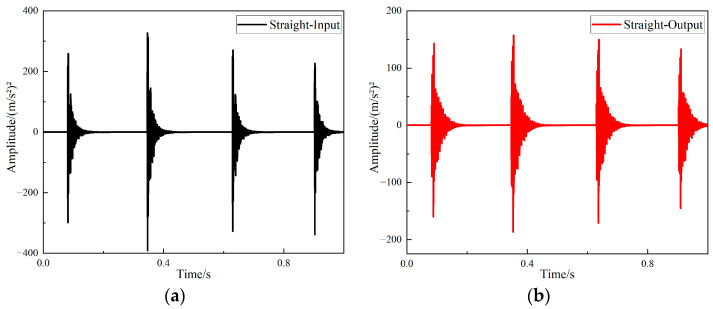
The response curves of the straight-tube structure support (**a**) Time domain input signal of straight-tube structure support, (**b**) time domain output signal of straight-tube structure support.

**Figure 15 materials-15-04308-f015:**
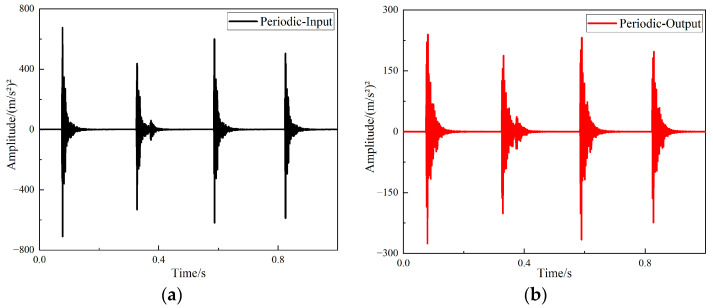
The response curves of the periodic structure support (**a**) Time domain input signal of periodic structure support, (**b**) time domain output signal of periodic structure support.

**Figure 16 materials-15-04308-f016:**
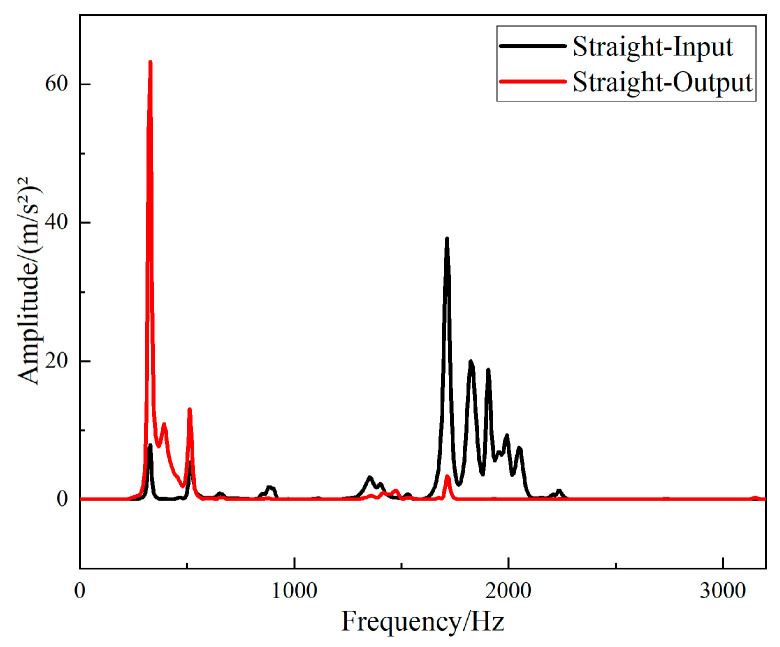
Frequency domain input and output signal of the straight-tube structure support.

**Figure 17 materials-15-04308-f017:**
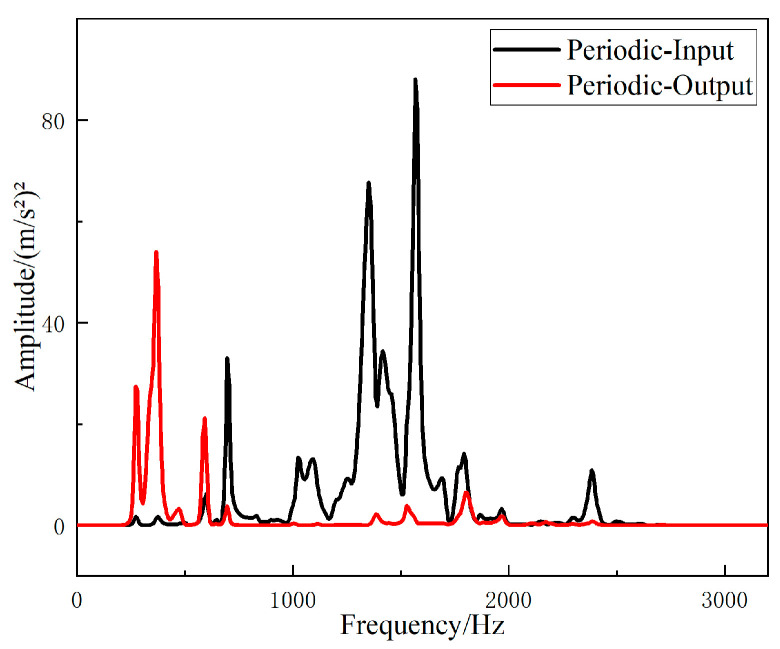
Frequency domain input and output signal of the periodic structure support.

**Figure 18 materials-15-04308-f018:**
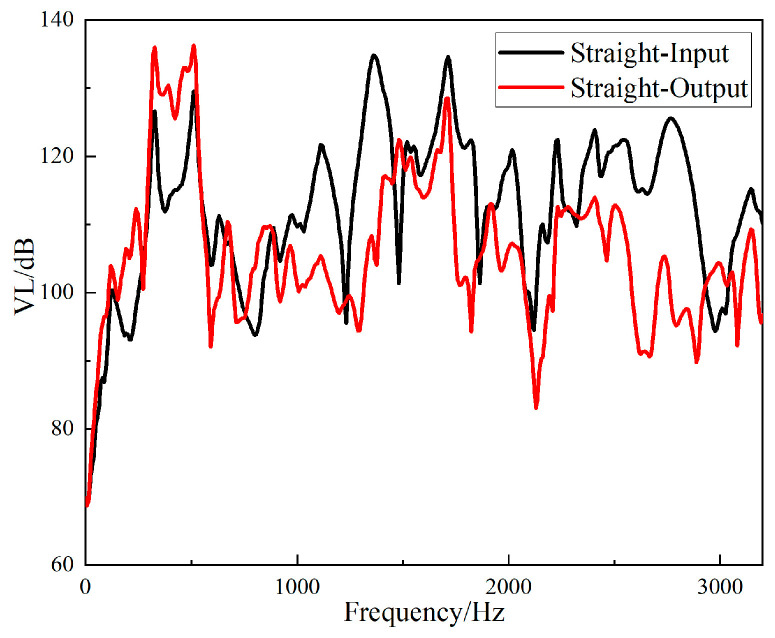
Frequency domain input and output *VL_Z_* of the straight-tube structure support.

**Figure 19 materials-15-04308-f019:**
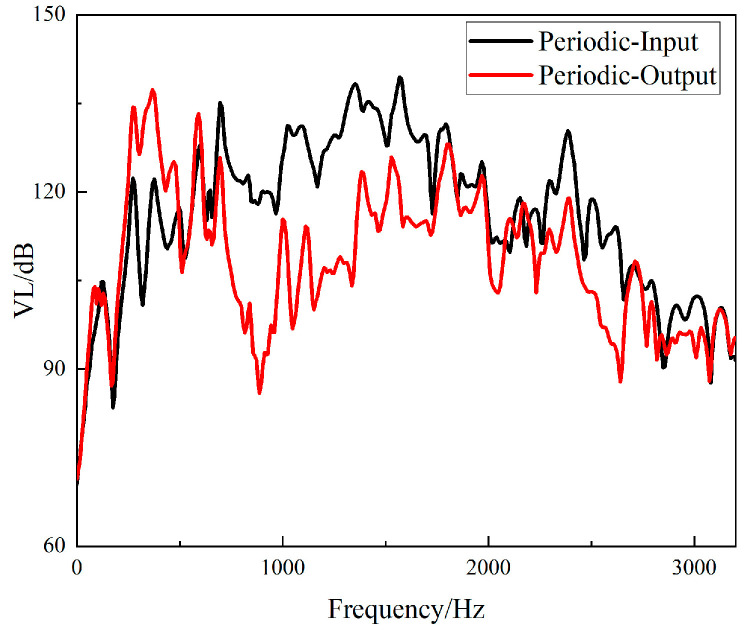
Frequency domain input and output *VL_Z_* of the periodic structure support.

**Figure 20 materials-15-04308-f020:**
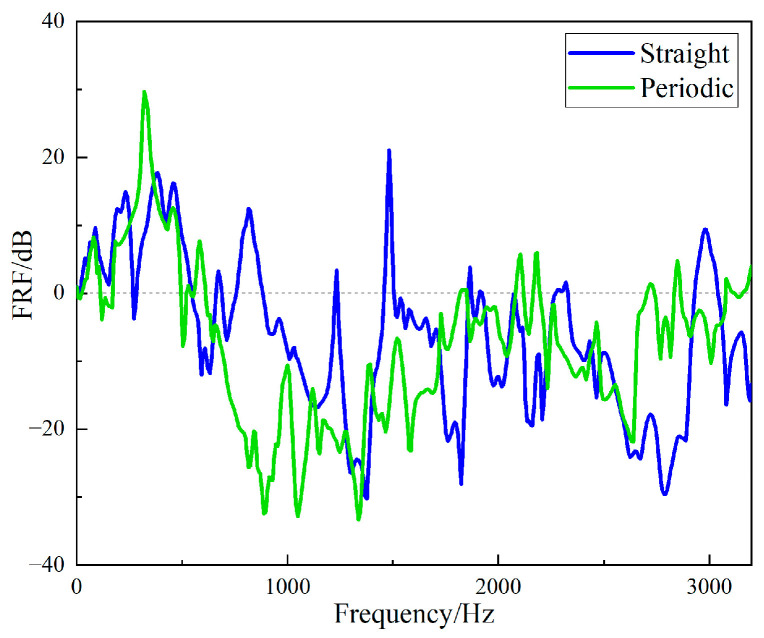
Frequency domains *FRF* of the two different specimens.

**Figure 21 materials-15-04308-f021:**
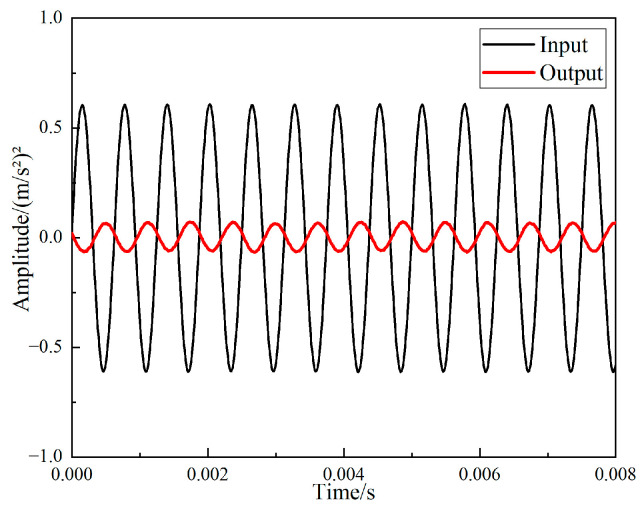
Time domain input and output signal of periodic structure support.

**Figure 22 materials-15-04308-f022:**
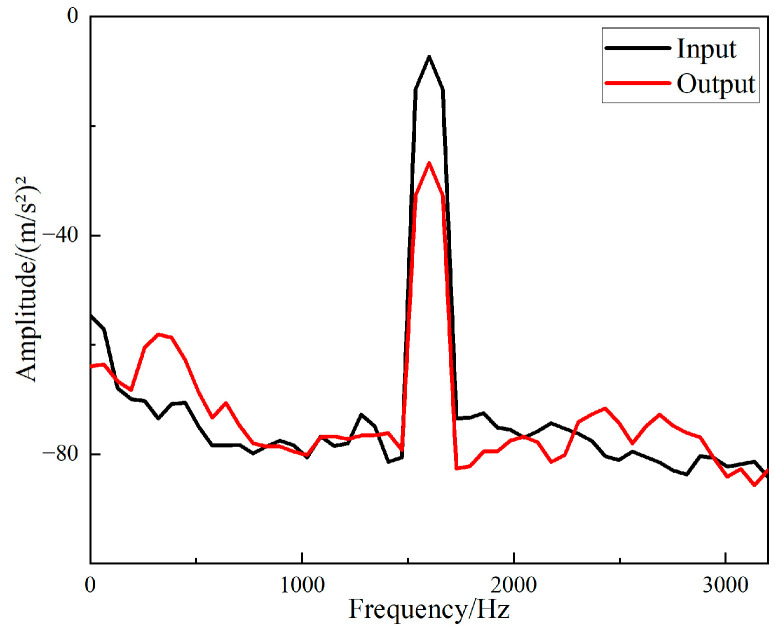
Frequency domain input and output signal of periodic structure support.

**Figure 23 materials-15-04308-f023:**
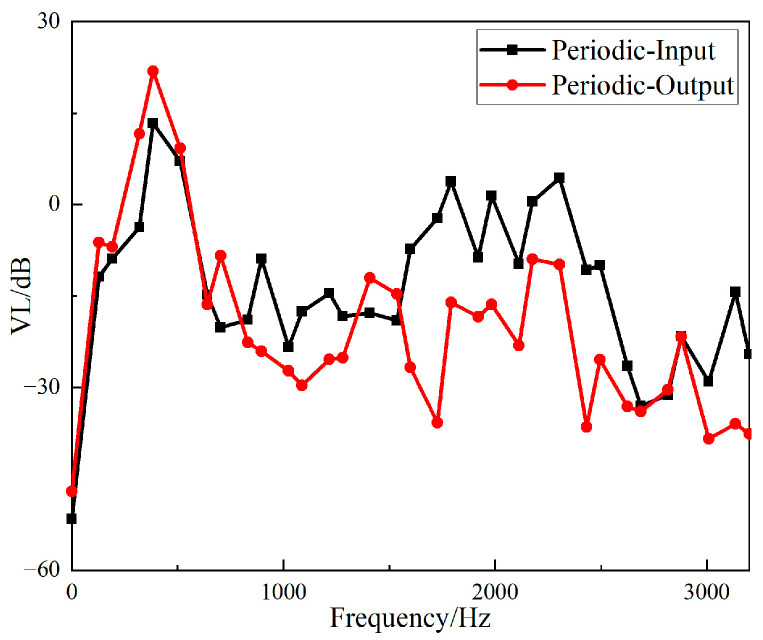
Frequency domain input and output *VLz* of the periodic structure support.

**Figure 24 materials-15-04308-f024:**
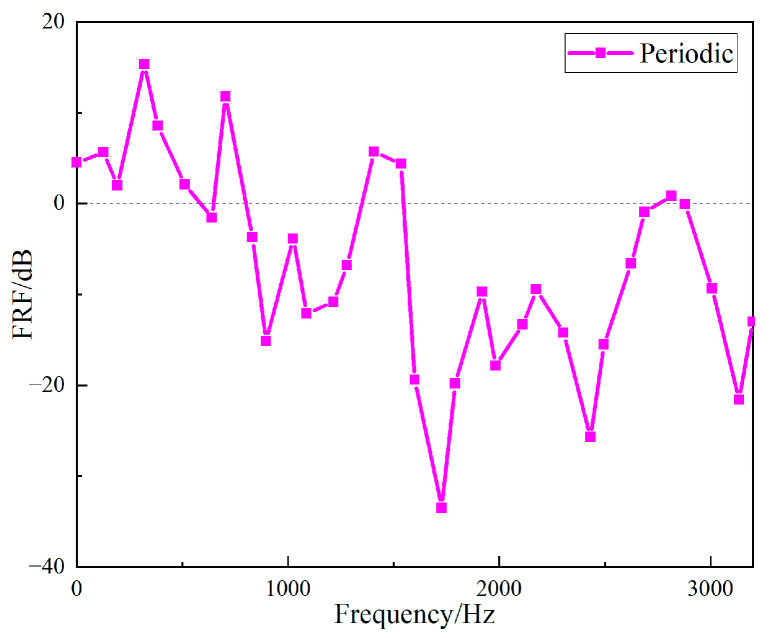
The Frequency domain *FRF* of the periodic structure support.

**Figure 25 materials-15-04308-f025:**
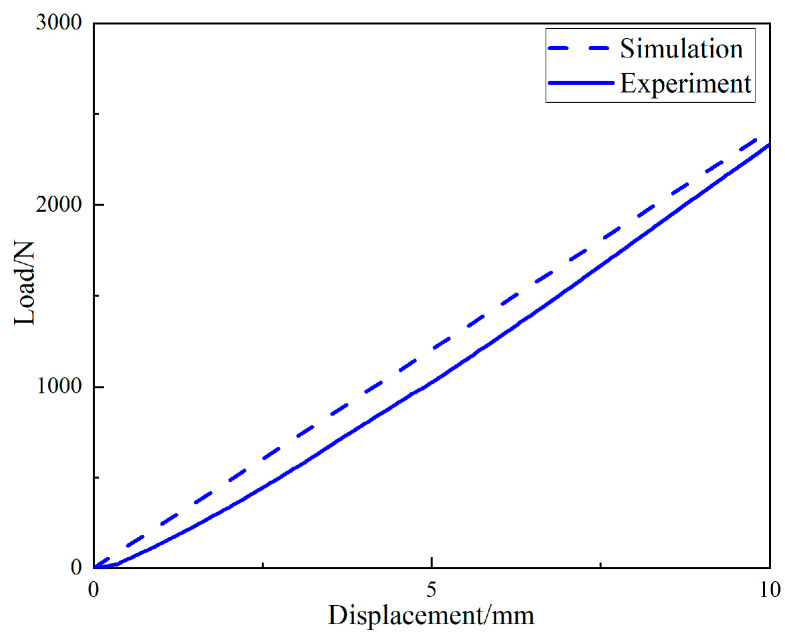
Comparison of load–displacement curve between simulation and experiment.

**Figure 26 materials-15-04308-f026:**
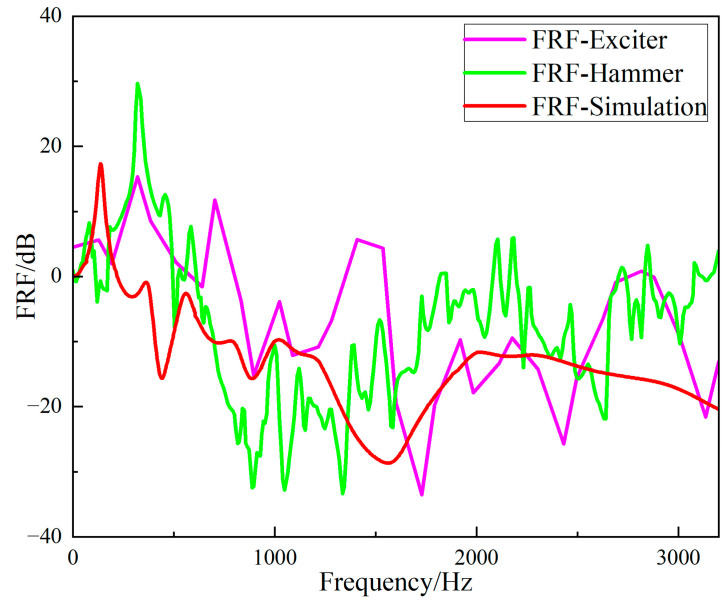
Comparison of *FRF* values between simulation and experiment.

**Table 1 materials-15-04308-t001:** Material parameters of specimens.

Material	Young’s Modulus (MPa)	Poisson’s Ratio	Density (g/cm^3^)
Acrylonitrile-butadiene-styrene	1098	0.33	1.13

**Table 2 materials-15-04308-t002:** The parameters of the experiment equipment.

Equipment	Model	Parameters
Piezoelectric accelerometer	4507Bx	9.688 mV/(m/s^2^)
Force hammer (Force sensor)	8206-002	2.27 mV/N
Data acquisition system	3660-C-LAN-XI	-
Analyzing software	BK Connect	-

## Data Availability

The data presented in this study are available on request from the corresponding author.

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
