# Peer review of "Investigations on the Band-Gap Characteristics of Variable Cross-Section Periodic Structure Support Made of Acrylonitrile-Butadiene-Styrene"

_materials, 2022, doi:10.3390/ma15124308_

Round 1

Reviewer 1 Report

I did not find measurement parameters such as sampling frequency, used filters, etc. in the text.
A hammer with a built-in force sensor was used for the measurement, but I did not find its use in data processing. I assume that in Fig. 7 the curves are drawn at the sensor near the impact (input) and on the opposite side (output). Why are the spectra of both samples not compared?
How many samples were tested?
Fig. 10 shows the excitation by the shaker. What signal was generated during the excitation - harmonic, random?

Is it possible to compare the characteristics of Fig. 7 a, b and Fig. 10?

What was the accuracy?

Reviewer 2 Report

This paper aims to study the band-gap characteristics of variable cross-section periodic structure supports that are made of ABS. The writing style of the article is acceptable and the proposed model of structure support was said to be efficient. However, in my opinion, the paper suffers more from the novelty part. In other words, it is much better to consider at least the following two parts for enriching the paper:

  1. Verify the performance of the proposed support model in a finite element model and try to develop a force-deformation relationship for this support.
  2. Next, consider other dimensions for Di, Li, and other parameters in figure 3 and try to develop rules that relate the efficiency of the support to those shape parameters.

In Figure1, the notation of stiffness and damping must be replaced with each other.

This paper can not be published in its current form but can be reconsidered for another review after completely addressing the mentioned comments.

Reviewer 3 Report

Comments are given below:

It's an interesting topic, and there are a few cases that seem interesting to write for manuscript editing. Therefore, we are unable to referee your manuscript at this time unless a general review is made and the manuscript is submitted.

1- The manuscript has a long introduction and is not well organized according to the subject of the manuscript.

2- The problem theory is not well stated and none of the equations presented have a reference.

3- The structure of the manuscript needs general corrections and according to the present study, the structure of the manuscript should include abstract, introduction, problem theory, experiments, numerical analysis, validation and verification, results and discussion, conclusion. For example, experimental results in sections of 3.3 and 4.2 and test method in sections of 3.2 and 4.1 were repeated twice.

4- In each novel research, the minimum number of input parameters and output results are more than a few items that are not observed in this manuscript.

5- The quality of the drawn graphs is not high and the writings inside the figures are not clear.

6- Use high quality images for setup images and provide a complete description about them.

7- The manuscript is more of a report, the references and standards are not mentioned. In addition, the conclusion section expresses the state of a simple result and has no specific innovations.

Reviewer 4 Report

Please provide more details in text about Fig.7.

Please provide more details in text about Fig.10.

Conclusion:"The research shows that compared with the straight-tube structure support, the periodic structure support proposed in this paper has more obvious vibration attenuation effect on the excitation from 688Hz to 1800Hz. In addition, the periodic structure support is stimulated at different frequencies by the shaker, and the band gap was judged to be from 800Hz to 2000Hz." Please provide more details in conclusion.

Reviewer 5 Report

The papers uses the band gap theory of periodic structure, a new variable cross-section periodic

structure support based on ABS is designed, and to measure the band gap by hammer method and shaker excitation. The paper is in good shape for publication. However, the following comments are required.

  • Please don’t mention an abbreviation in the title. It is not defined what ABS is yet in the title.
  • Again, in the abstract line 2, the ABS is not yet defined.
  • Line 113, it is not appropriate to mention “scholars from various countries”.
  • Line 131, it is not appropriate to mention what university the researcher is coming from.
  • This paper needs more discussion about the experimental results.

Round 2

Reviewer 2 Report

Most of my previous comments were well answered. This paper can be published after preparing the text within an appropriate format and based on the editor's decision.

Reviewer 3 Report

Comments are given below:

Thanks for the consideration and the revised manuscript. I think it is much improved.

1. The quality of most Figures is not suitable and they need to be corrected. Use a variety of high-quality colors, especially Figures that have graphs.

2. Put clearer pictures of the test setup in Figures 5 and 6 and explain more about their function in the manuscript.

3. In my opinion, the word ABS is mostly used for anti-lock brakes in the car, so it is suggested to use the main words in the whole manuscript.

4. How to trust numerical results without validating the results. Therefore, in order to improve the work, it is necessary to validate the results.

5. After final editing, the manuscript needs a structure arrangement that is not already done.

6. Due to the importance of the numerical method in the case study, the following refs may help the numerical method converge for validation.

a) R. Masoudi Nejad, P. Noroozian Rizi, M.S. Zoei, K. Aliakbari, H. Ghasemi, Failure Analysis of a Working Roll Under the Influence of the Stress Field Due to Hot Rolling Process, J. Fail. Anal. Prev. (2021). https://doi.org/10.1007/s11668-021-01131-9.

b) K. Aliakbari, R. Masoudi Nejad, T. Akbarpour Mamaghani, P. Pouryamout, H. Rahimi Asiabaraki, Failure analysis of ductile iron crankshaft in compact pickup truck diesel engine, Structures. 36 (2022) 482–492.

Reviewer 5 Report

Thank you for considering my comments.
